# Sub-optical-cycle light-matter energy transfer in molecular vibrational spectroscopy

**Martin T. Peschel** [1,5], **Maximilian Högner** [2,3,5], **Theresa Buberl** [2,3,5], **Daniel Keefer** [1,4], **Regina de Vivie-Riedle** [1] ✉ **& Ioachim Pupeza** [2,3] ✉

The evolution of ultrafast-laser technology has steadily advanced the level of detail in studies of light-matter interactions. Here, we employ electric-field-resolved spectroscopy and quantum-chemical modelling to precisely measure and describe the complete coherent energy transfer between octave-spanning mid-infrared waveforms and vibrating molecules in aqueous solution. The sub-optical-cycle temporal resolution of our technique reveals alternating absorption and (stimulated) emission on a few-femtosecond time scale. This behaviour can only be captured when effects beyond the rotating wave approximation are considered. At a femtosecond-to-picosecond timescale, optical-phase-dependent coherent transients and the dephasing of the vibrations of resonantly excited methylsulfonylmethane (DMSO$_2$) are observed. Ab initio modelling using density functional theory traces these dynamics back to molecular-scale sample properties, in particular vibrational frequencies and transition dipoles, as well as their fluctuation due to the motion of DMSO$_2$ through varying solvent environments. Future extension of our study to nonlinear interrogation of higher-order susceptibilities is fathomable with state-of-the-art lasers.

Static or vibrationally-induced asymmetric charges in molecules cause electric dipole moments, responsible for efficient coupling to infrared (IR) radiation[1]. Optical energy transferred from an excitation IR field to vibrating molecules can either dissipate incoherently in the form of heat, or can be re-emitted with a fixed phase relation to the excitation field, i.e., coherently. Traditional frequency-domain vibrational spectroscopy methods such as direct-absorption spectroscopy or Fourier-transform spectroscopy provide wavelength-resolved absorbance (and phase) information, obtained via temporal integration over the duration of the interaction[2–4]. Figure 1a illustrates the result of such a measurement for a Lorentzian-shaped absorption line which is typical for linearly-interrogated, homogeneously-broadened resonances of molecules embedded in a solvent[5].

IR molecular absorption spectra contain rich information about molecular composition, abundance and conformation, making

vibrational spectroscopy a widely-applied tool in fields including fundamental science[1,6], analytical chemistry[7], and the life sciences[2,8]. However, in traditional frequency-domain spectroscopy, temporal integration hides the transient energy transfer between the light field and the material sample, obscuring deeper insight into the underlying dynamic light-matter interaction.

This work presents a quantitative study of the complete dynamics of the coherent energy transfer between broadband mid-IR optical waveforms and vibrating molecules in aqueous solution, with sub-optical-cycle temporal resolution. Field-resolved IR spectroscopy[9] permits the differentiation between the qualitatively different transient energy transfer for few-cycle excitation (FCE) and chirped-pulse excitation (CPE). In both cases, sub-cycle resolution allows us to observe ultrafast dynamics caused by effects beyond the rotating wave approximation (RWA). Furthermore, FCE allows for temporal

[1]Ludwig-Maximilians-Universität München, Butenandtstraße 5-13, 81377 Munich, Germany. [2]Max-Planck-Institut für Quantenoptik, Hans-Kopfermann-Straße 1, 85748 Garching, Germany. [3]Ludwig-Maximilians-Universität München, Am Coulombwall 1, 85748 Garching, Germany. [4]Department of Chemistry, University of California, Irvine, CA 92697, USA. [5]These authors contributed equally: Martin T. Peschel, Maximilian Högner, Theresa Buberl.
✉e-mail: regina.de_vivie@cup.uni-muenchen.de; ioachim.pupeza@mpq.mpg.de

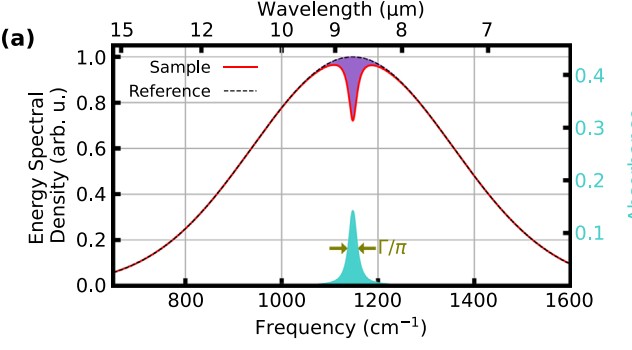

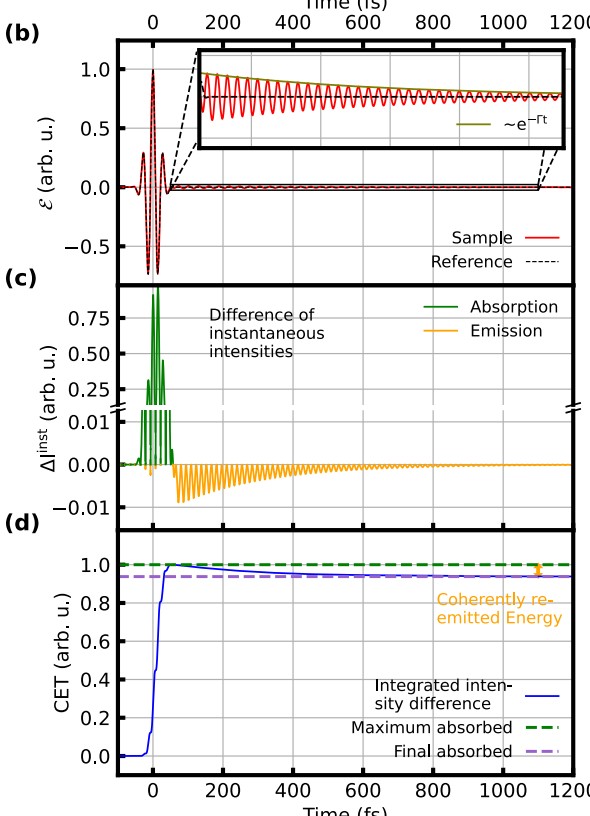

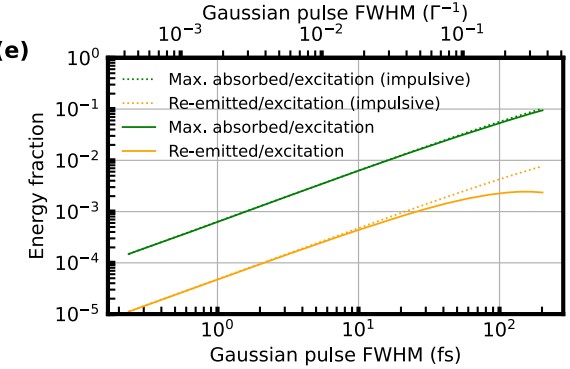

**Fig. 1 | Time-integrated and time-resolved broadband vibrational spectroscopy. a** Model spectrum before (black, dashed) and after (red) transmission through a Lorentzian absorber centered at 1148 cm$^{-1}$ with a full-width-at-half-maximum (FWHM, $\Gamma/\pi$) of 19.4 cm$^{-1}$, shown for conventional (time-integrating) absorption spectroscopy. Purple shaded area: difference between the two spectra, amounting to the energy incoherently dissipated in the molecular sample. Turquoise shaded area: normalized absorbance, defined as the difference between the logarithms of the two spectra log$_{10}$(|Reference|)−log$_{10}$(|Sample|). **b** Electric field $\mathcal{E}(t)$ of the 30-fs-full-width-half-maximum Gaussian excitation pulse (black, dashed) with the spectrum shown in black in **a**. Red: coherent response of the Lorentzian absorber shown in **a** to this few-cycle excitation (FCE). Olive: decay of the molecular response due to dephasing proportional to $e^{-\Gamma t}$. **c** Time-resolved difference $\Delta I^{\text{inst}}(t)$ of the instantaneous intensities of the electric fields in **b**. The positive values (green) correspond to absorption, the negative values (yellow) to coherent emission. For FCE, the coherent emission event has no contribution from the excitation. **d** Integration of $\Delta I^{\text{inst}}(t)$ shown in **c** yields the coherent energy transfer CET$(t)$. Green dashed line: maximum of CET$(t)$, i.e., the maximum transiently absorbed energy. Purple dashed line: end value of CET$(t)$, equal to the purple shaded area in **a**. Orange double arrow: difference between maximum and end value, corresponding to the energy which is coherently re-emitted by the Lorentzian absorber. **e** Ratio of the maximum transiently absorbed energy (green) and the coherently-re-emitted energy (yellow) to the excitation pulse energy, as a function of the FCE-pulse duration. These ratios deviate by less than 30% from the values for the impulsive excitation regime (dotted lines, see also Supplementary Information) for pulse durations below one-tenth of the typical Lorentzian decay time $\Gamma^{-1}$ (here, 548 fs).

sample susceptibility, which is determined by the microscopic vibrational transition dipole moments, eigenfrequencies, and dephasing times of the vibrational modes[13–16].

## Results and discussion
### Energy balance in the impulsive regime
We define absorption and coherent emission as events of energy transfer from the impinging coherent light beam to the molecules and vice versa. For a FCE (Fig. 1b), the events of absorption and coherent emission (in analogy to the nuclear magnetic resonance phenomenon of free induction decay[10]) are separated in time to a large extent (Fig. 1c, d). While the absorption event is governed by an interference between the excitation waveform and the molecular response, the coherent emission consists only of the latter (after the decay of the excitation). The brevity of the absorption event implies little spectral specificity. In fact, for an impulsive excitation, the absorption information reduces to a frequency-independent scalar equal to the total oscillator strength of the sample in the spectral region covered by the excitation. In contrast, the emitted field after the FCE contains the frequency-dependent sample-specific spectroscopic information. The ratio of its integrated energy to the excitation energy (Fig. 1e, yellow) is indicative of the efficiency with which spectroscopic information can be extracted from the sample. It depends on the environment-dependent transition dipole moments and dephasing times of the molecular sample. The ratios of the maximum absorbed as well as coherently re-emitted energy to the excitation energy are plotted in Fig. 1e as functions of the duration of a Gaussian FCE, showing that FCEs can be regarded as impulsive for pulse durations below one-tenth of the typical Lorentzian decay time (in our experiment, hundreds of femtoseconds).

### Few cycle excitation (FCE)
In our experiment, the 1-µm pulses of an Yb:YAG thin-disk oscillator were compressed to 15 fs, subsequently driving intrapulse difference-frequency generation in a nonlinear crystal. This resulted in waveform-stable IR pulses covering the 850-to-1670-cm$^{-1}$ spectral range, which were transmitted through a 30-µm liquid cuvette containing either pure water (reference measurement) or a 10-mg/ml solution of methylsulfonylmethane (DMSO$_2$) in water (sample measurement).

After transmission through the liquid cuvette, the waveforms were recorded via electro-optic sampling[9] (see also Supplementary

separation between absorption and emission, two processes that are qualitatively different in that only coherent re-emission is affected by solvation dynamics. For CPE, additional coherent transients emerge[10–12]. These are caused by the interference between the resonant system response and the non-resonant part of the chirped light wave, which causes additional series of absorption and emission. The discussed effects are quantitatively described by the macroscopic

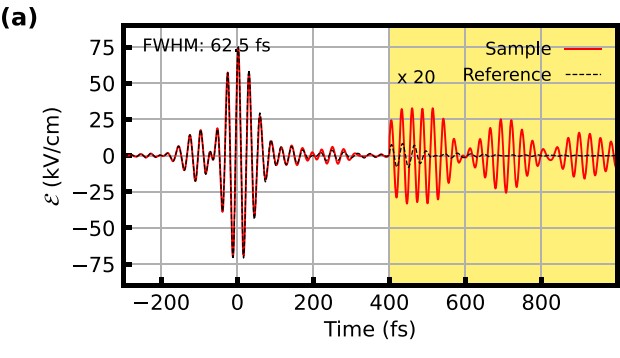

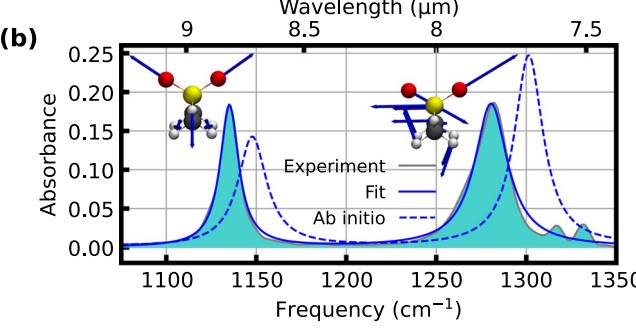

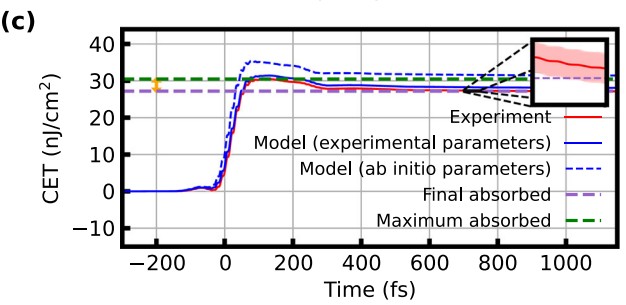

**Fig. 2 | Few-cycle excitation (FCE). a** Electric field of the compressed excitation pulse after transmission through a 30-μm thick layer of water (black, dashed) and a solution of 10 mg/ml DMSO₂ in water (red). **b** Absorbance derived from the measurements in **a**. The molecular structures illustrate the corresponding displacement vectors (blue arrows) of the vibrational modes responsible for the two main absorption bands, the symmetric (left) and asymmetric (right) SO₂ stretching vibrations. Solid blue: Lorentzian fits, dashed blue: ab initio calculations (see text). **c** Coherent energy transfer CET(t) between the excitation field and the DMSO₂ sample, extracted from the experimental data (solid red line: mean value of three consecutive measurements, each comprising 20 scans, shaded red area: error corridor determined with Gaussian error propagation, see Methods), from modeling with the fitted Lorentzian parameters (solid blue line), and with Lorentzian parameters obtained ab initio (dashed blue line). The dashed green and purple lines show the maximum and end level of the CET, respectively. Their difference is the coherently emitted energy, which is 11% of the maximum absorbed energy (orange double arrow).

Information). In the first experiment, the reference waveform was compressed to a duration of 62.5 fs (Fig. 2a). Figure 2b shows the corresponding sample absorbance, dominated by the resonances of the asymmetric and symmetric stretching vibrations of the SO₂ group.

Temporal integration of the instantaneous intensity difference between reference and sample signals from a time instant $t_0$ well before the pulse until the (variable) instant $t$ yields the coherent energy transfer between sample and light field, CET(t) (Fig. 2c):

$$\text{CET}(t) = \int_{t_0}^{t} \left( I_{\text{ref}}^{\text{inst}}(t') - I_{\text{sam}}^{\text{inst}}(t') \right) dt', \qquad (1)$$

where $I_{\text{ref,sam}}^{\text{inst}}(t) = c\varepsilon_0 |\mathcal{E}_{\text{ref,sam}}(t)|^2$ is the instantaneous intensity, defined as the magnitude of the Poynting vector, with the real electric field $\mathcal{E}_{\text{ref,sam}}(t)$. This macroscopic quantity is identical to the microscopic coherent energy transfer between the light field and individual molecules, if the reference measurement is taken as the excitation waveform instead of the actual, propagation-dependent electric field (see Supplementary Information). The level asymptotically approached by CET(t) for $t \to \infty$ corresponds to the integrated difference of the energy spectral density (ESD) measured in frequency-domain spectroscopy (cf. purple shaded area in Fig. 2b). In our measurement, around 11% of the absorbed energy (0.4% of the impinging energy) is coherently re-emitted (Fig. 2c). For a laser pulse described by a Dirac delta function, we obtained a value of 10% using the ab initio numerical model (see Supplementary Information), confirming that our experimental conditions are close to the impulsive excitation regime.

### Chirped pulse excitation (CPE)

In a second experiment, we used a 5-mm-thick CaF₂ substrate in the IR beam path to chirp the excitation pulse (Fig. 3a). This leads to a longer pulse duration, such that in contrast to the FCE, a significant portion of the coherent emission now overlaps temporally with the excitation pulse. While the time-integrated ESD (Fig. 3b) yields absorbance information equivalent to that obtained in the FCE case, evaluating the coherent energy transfer with Eq. 1 reveals oscillations in CET(t) which are not observed in the case of the FCE (Fig. 3c). The maxima of these oscillations, visible in the ab initio model calculations around 420 fs and 720 fs, can be attributed to the symmetric stretching vibration and the one around −180 fs to the asymmetric stretching vibration. They occur in the experiment around 80 fs later since to the ab initio calculations are blue-shifted with respect to the experiment. Our model (see below) allows us to only include certain vibrational modes in the simulation, thus enabling us to clearly identify certain features with corresponding vibrational modes. The first oscillation maxima are clearly visible, the later maxima of the asymmetric stretch overlap with the signal due to the symmetric stretch. The oscillations are damped by dephasing and by the pulse intensity decreasing over time.

The emerging alternating sequence of absorption and coherent emission, also known as coherent transients[17], was previously observed for atomic transitions in the visible spectral range[11,12,18,19]. It is caused by the fact that after the resonant transition, the phase of the chirped pulse shifts with respect to the phase of the oscillating system. Whether emission or absorption occurs, depends on the current phase relationship. In addition, there is a second effect that is revealed by a more detailed look at CET(t) (Fig. 3d). Underlying sub-optical-cycle dynamics can be observed: Energy is absorbed or emitted in a stepwise fashion with each half-cycle of the electric field.

### Theoretical model

To quantitatively explain the energy transfer dynamics observed in the experiment, we built a model based on time-dependent first-order perturbation theory. Under the assumption of a linear sample response with respect to the impinging power (which is experimentally confirmed within the precision of our measurement[20]), according to the optical Bloch equations the susceptibility $\chi^{(1)}(\omega)$ of a molecular solution of concentration $\beta$ of a molecule of mass $m$ is given by[6,21]:

$$\chi^{(1)}(\omega) = \sum_k \frac{2}{3} \frac{\beta}{mh\varepsilon_0} \frac{\mu_{0k}^2 \omega_k}{\omega_k^2 - (\omega - i\Gamma_k)^2}. \qquad (2)$$

Thus, in the spectral range covered by our laser field $\mathcal{E}(t)$, each vibrational mode $k$ forms a Lorentz oscillator, parametrized by three parameters: the central frequency $\omega_k$, the transition dipole moment $\mu_{0k}$ and the homogeneous dephasing rate $\Gamma_k$. We consider all 27 vibrational modes of DMSO₂. The parameters $\omega_k$, $\mu_{0k}$ and $\Gamma_k$ can either

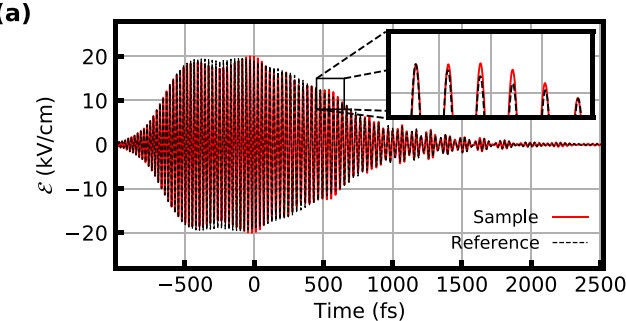

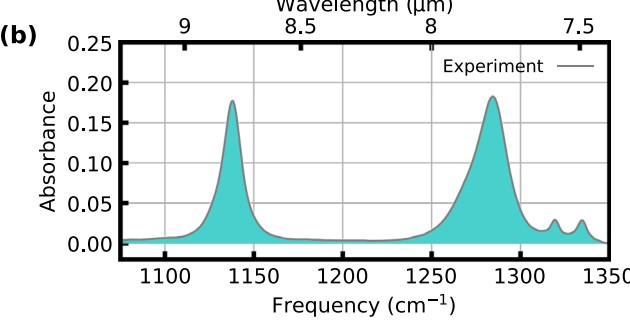

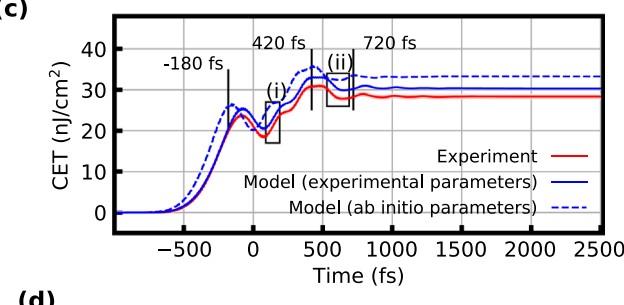

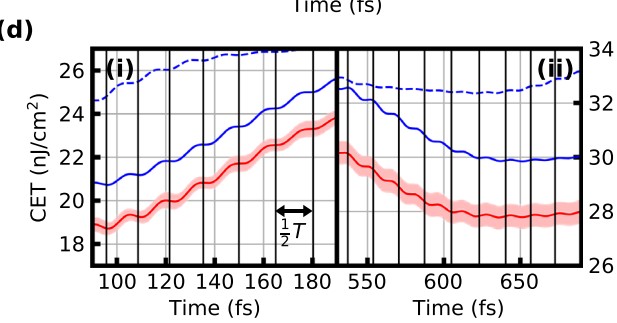

**Fig. 3 | Chirped-pulse excitation (CPE). a** Electric field of the chirped excitation pulse after transmission through a 30-μm thick layer of water (black, dashed) and 10-mg/ml solution of DMSO$_2$ in water (red). **b** Absorbance, calculated from the measurements in **a**. **c** Coherent energy transfer CET($t$) between the excitation field and the DMSO$_2$ solution, extracted from the experimental data (solid red line: mean value of three consecutive measurements, each comprising 20 traces, shaded red area: error corridor determined with Gaussian error propagation, see Methods), from modeling with the fitted Lorentzian parameters (solid blue line, see Fig. 2), and with Lorentzian parameters obtained ab initio (dashed blue line). The maxima of CET($t$) around −180 fs, 420 fs, and 720 fs are marked by vertical black lines (see also the discussion in the main text and Fig. 4d). **d** Magnified displays of the sections encased by the dashed black rectangles (i) and (ii) in **c**, revealing the sub-optical-cycle structure of absorption and stimulated emission. The zeros of the electric field are marked by vertical black lines, showing that each absorption or emission step corresponds to one half-cycle ($\frac{1}{2}T$) of the electric field.

be fitted to the experimental data, or they can be determined by ab initio quantum-chemical calculations. The simulation curves shown in Fig. 2b, c and Fig. 3c, d were obtained from the total susceptibility as described in the Supplementary Information.

In a time-domain description, the interaction of the laser field $\mathcal{E}(t)$ with each vibrational mode leads to a coherence[6]

$$\rho_{0k}^{(1)}(t) = \frac{i}{\hbar}\mu_{0k}\int_0^\infty \mathcal{E}(t-t_1)e^{i\omega_k t_1}e^{-\Gamma_k t_1}dt_1,\qquad(3)$$

which causes a polarization

$$P_k^{(1)}(t) = F^{-1}\{\varepsilon_0 \cdot \chi_k^{(1)}(\omega) \cdot \mathcal{E}(\omega)\}(t) = \frac{2}{3}\frac{\beta N_A}{M}\mu_{0k}\mathrm{Re}\left(\rho_{0k}^{(1)}\right).\qquad(4)$$

The fields thus created from each infinitesimal sample volume (DMSO$_2$ molecules and their immediate solvent environment) interfere with each other and with the incident field to create the observed response in the transverse mode of the laser.

### Sub-optical-cycle dynamics

The curves in Fig. 4a–c show the evolution of the coherence of one exemplary vibrational mode, the symmetric stretching vibration, in the complex plane, calculated using Eq. 3. One may invoke the RWA[22], which neglects terms oscillating with the sum of the vibrational and field frequencies to obtain the curves in Fig. 4a, c. However, field-resolved spectroscopy resolves sub-optical-cycle dynamics, directly revealing the influence of these terms. It is evident comparing Fig. 4a–d that the sub-cycle dynamics vanish under the RWA. The term neglected in the RWA imprints a cycloid structure onto the polarization, which corresponds to the step-like patterns and sub-cycle oscillations in Fig. 3d.

### Coherent transients

The coherent transients observed with CPE are illustrated by the spiral shapes in Fig. 4c, d, when considering that CET($t$) in Fig. 3c is proportional to the squared absolute value of the coherence (see Supplementary Information). The system starts at $t = t_0$ in equilibrium at the origin. Then, the chirped pulse interacts with the system. At first, this interaction is off-resonant, which leads to a spiraling around the origin with increasing amplitude and decreasing frequency, as resonance is approached. This spiral corresponds to a slow increase of CET($t$) in Fig. 3c at early times. At resonance, a large increase in the magnitude of the coherence is observed. Finally, the off-resonant interaction once more leads to a spiral pattern, which manifests as the damped oscillation in Fig. 3c at later times, as the magnitude of the polarization increases and decreases with each loop. Thus, the contributions of the non-resonant excitation to the complex amplitude cause the sequence of absorption and stimulated emission. Figure 4d also illustrates that the maxima in Fig. 3c around 420 fs and 720 fs are caused by the symmetric stretching vibration. The marked time points correspond to maxima in the squared absolute value of the coherence.

### Ab initio quantum chemical calculations

The ab initio model permits an in-depth study of the influence of the surrounding solvent on the vibrating molecules. The transition frequency distribution of the DMSO$_2$-water clusters demonstrates that the observed dephasing is mainly due to the dynamics of the molecule in the solvent (homogeneous broadening) and not due to their static structure (inhomogeneous broadening). This agrees with the experimentally observed Lorentzian line shapes, which can also be observed in ab initio molecular dynamics simulations (see Methods section and Supplementary Figs. 3 and 4).

The calculated frequencies and transition dipole moments for the different models are presented in Table 1. Due to solvation, absorption maxima are red-shifted, and transition dipole moments increase significantly. As a result of the latter, the absorbed and coherently re-emitted fractions of the impinging energy are larger in the polarizable continuum model (PCM, see Fig. 4e). It must be

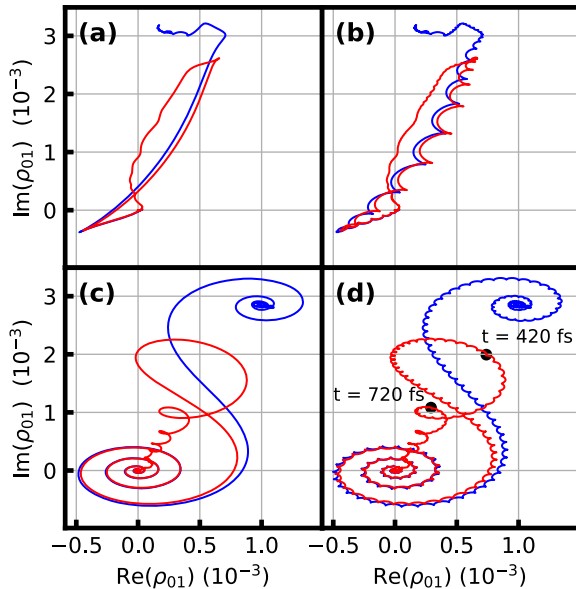

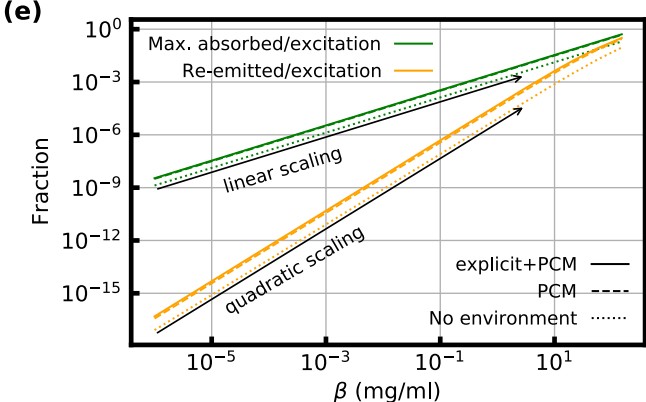

**Fig. 4 | Calculated coherence and energy transfer ratios.** Calculated vibrational molecular coherence in the symmetric stretching vibrational mode of the $DMSO_2$ molecule in solution displayed in a frame rotating with the vibrational eigenfrequency. The calculations are done with **a** the compressed pulse using the RWA, **b** the compressed pulse without the RWA, **c** the chirped pulse using the RWA, and **d** the chirped pulse without the RWA. The red curves include relaxation; the blue curves do not include relaxation. Each bump of the cycloid in **b** corresponds to one half-cycle of the electric field, showing that energy transfer from the excitation field to the molecular system is completed after 3 to 4 field cycles (Fig. 2b) in the case of the FCE. In **d**, the time points that are maxima of CET($t$) caused by the symmetric stretching vibration are marked with black dots. **e** Re-emitted (orange) and maximum absorbed (green) fraction of the impinging pulse energy versus concentration. The results were obtained from the impulsive-regime model (see Supplementary Information) with the FCE and ab initio Lorentz parameters. The solid lines include direct interactions with the surrounding water and the screening effect of the polarizable continuum, the dashed lines only the latter, and the dotted lines neither. For small concentrations, the maximum absorbed energy scales linearly with concentration. In contrast, the coherent re-emission, containing the spectroscopic information, scales quadratically with the concentration.

noted that these fractions also depend on the concentration of the sample molecules. For small concentrations, the maximum absorbed energy scales linearly with the concentration. The coherent re-emission, containing the spectroscopic information, scales quadratically with the concentration due to the absence of homodyning with the excitation field (see Supplementary Information). This scaling behavior is in line with the Beer–Lambert law and derives from the linear relationship between the concentration and the amplitude difference caused by the sample.

**Table 1 | Parameters characterizing the resonances of $DMSO_2$**

| | $\tilde{v}_{01}$ (cm$^{-1}$) | | $\mu_{01}$ (D) | |
|---|---|---|---|---|
| | Symmetric | Asymmetric | Symmetric | Asymmetric |
| Calculation (no environment) | 1192 | 1311 | 0.24 | 0.26 |
| Calculation (explicit solvation) | 1148 | 1301 | 0.33 | 0.41 |
| Calculation (PCM) | 1142 | 1288 | 0.32 | 0.37 |
| Experiment | 1135 | 1280 | 0.30 | 0.38 |

Calculated and experimental absorption wavenumbers $\tilde{v}_{01}$ and transition dipole moments $\mu_{01}$ for the symmetric and asymmetric stretching vibration of $DMSO_2$, which are the main modes contributing to the observed spectral range (Fig. 2b). The calculations used the M06-2X density functional[13] and were performed on a single molecule without an environment, a single molecule embedded in a PCM or using explicit solvation. In the latter case, different $DMSO_2$-water clusters were embedded in a PCM. The experimental values were obtained by fitting Lorentzians to the data presented in Fig. 2b.

To explain the increase of the transition dipole moments due to the PCM, we examined partial charges of the oxygen atoms in the density-functional theory simulations. This shows that the environment reduces the electrostatic interaction between different parts of the molecule, leading to a more asymmetric charge distribution. This causes higher dipole moments and less rigid bonds. The static polarizability of the solvent environment thus enhances the molecule-field interaction. This effect is equally well described by both the explicit solvent model and the PCM. However, with the explicit solvent model the molecular dynamics of the entire system can be simulated, which provides qualitative estimates of the dephasing rate $\Gamma$ (see Eq. 2 and Supplementary Fig. 4).

In conclusion, sub-optical-cycle-temporal-resolution field-resolved spectroscopy together with ab initio modeling provides quantitative, molecular-scale insight into the complete dynamics of the coherent energy transfer between broadband infrared light and vibrating molecules in solution. For an ultra-brief, broadband excitation, time-domain detection allows for the distinction between two qualitatively different light-matter energy transfer regimes: Absorption is governed by the strengths of the transition dipole moments for each individual molecule, independently. The generation of vibrational coherence with each half-cycle of the electric field is resolved. Emission yields a coherent sample-specific spectroscopic fingerprint carrying dephasing information due to molecular motion in varying environments. For spectroscopic techniques that allow for the separation of these two regimes in time[9,23,24] or in space[25,26], the ratio of the coherently transferred optical energy to the excitation energy (Fig. 4e) provides an indicator for the required spectroscopic detection sensitivity. For example, for our test molecule $DMSO_2$ at physiologically relevant[27] sub-ng/ml concentrations in water, the energy coherently re-emitted as a spectroscopic fingerprint amounts to $<10^{-16}$ of the excitation energy (and to $<10^{-8}$ of the absorbed energy), providing quantitative guidelines for the design of corresponding spectroscopic experiments. Our experimentally-validated ab initio model directly connects these energy transfer ratios to vibrational transition frequencies and dipole moments of the $DMSO_2$ molecules that are modulated by static and dynamic solvent interactions. We resolve sub-cycle effects beyond the RWA in the vibrational molecular coherence, and observe coherent transients in the context of field-resolved infrared spectroscopy.

The peak field strengths of less than 75 kV/cm in our experiment populate the first excited state only on the order of $10^{-3}$, rendering nonlinearities in the sample response negligible. With the availability of high-intensity, broadband, waveform-stable mid-IR sources[28–31] the extension of our study to field-resolved nonlinear (multidimensional) coherent spectroscopies[6,32–34] in the molecular fingerprint region becomes promising in the near future. This would enable studies of the coupling between vibrational modes and of the correlation functions

describing the fluctuating environments using phase-sensitive direct observation of the electrical field with unprecedented time resolution.

## Methods

### Field-resolved spectroscopy

The setup used for field-resolved spectroscopy (see also Supplementary Fig. 1) is similar to the one previously reported[9]. In short, all-solid-state spectral broadening via self-phase modulation in a two-stage multi-pass cell and subsequent compression driven by an Yb:YAG thin-disk oscillator operating at a repetition rate of 28 MHz serves as a high peak and average power source for the experiments. The resulting 15-fs output pulses centered at a wavelength of 1030 nm, and with an average power of 66 W generated, by frequency-down-converting, phase-stable mid-infrared (MIR) waveforms via intrapulse difference frequency mixing in a 1-mm thick $LiGaS_2$ crystal. After transmission of the super-octave MIR waveforms, covering the 850-to-1670-$cm^{-1}$ spectral range with an average power of 52 mW, through a 30-µm thick liquid cuvette (focus size 300 µm), a set of six customized dispersive mirrors is utilized to temporally compress their duration. The liquid cuvette is connected to a robotic liquid-handling system to automatically exchange the purified-water reference and the 10-mg/ml $DMSO_2$ solution.

The electric field of the MIR waveform emerging after the liquid cuvette is detected via electro-optic sampling (EOS) using a variably-delayed copy of the 15-fs, near-infrared driving pulse with an average power of 350 mW for gating. An 85-µm thick GaSe crystal under an angle of incidence of $\theta = 43°$ was used for EOS[9]. A mechanical chopper placed before the cuvette enables lock-in detection at a frequency of 7.5 kHz. The mutual delay between the MIR wave and sampling pulse was measured with interferometric delay tracking[35]. To avoid molecular background signal due to water vapor the system was operated at a pressure of $10^{-3}$ mbar from the MIR generation downstream. For the data shown in Figs. 2 and 3, the stage was scanned by 1.07 mm alternatingly in forward and backward direction, and 20 traces with a measurement time of 4.3 s each were averaged. This procedure was repeated three times to allow for quantification of the measurement error. The error bars for the normalized reference traces and their norm itself was estimated by the standard deviation of the three measurements. Each sample measurement was normalized to the average reference measurement, within the same spectral interval, to compensate for fluctuations in the MIR power. Normalization was done on the energy contained in the spectral interval 984–1051 $cm^{-1}$, where no significant molecular resonances are expected. Gaussian error propagation was used to compute the error of the CET.

### Quantum chemical calculations

The ground state geometry of $DMSO_2$ was obtained by geometry optimization with the M06-2X functional[13] and the 6−31 + g** basis set in the program package Gaussian16 (see Supplementary Table 1)[36]. Default convergence parameters were used. Subsequent harmonic frequency analysis showed no imaginary vibrational modes, thus confirming that the minimum energy structure had been found. Modes 14 and 15 were identified as the symmetric and asymmetric O-S-O stretching vibrations observed in the experiment (see Supplementary Tables 2 and 3).

To include anharmonicities in the description, an approach that explicitly calculates the potential energy surface for the vibrational modes 14 and 15 was chosen. Thus, single-point calculations were performed on $DMSO_2$ geometries that were displaced in steps along the two most relevant normal modes (see Supplementary Table 2, stepsizes: 0.2 Bohr for Mode 14, 0.1 Bohr for Mode 15) using M06-2X/6-31 g*, with and without a PCM as implemented in the program package Gaussian16. The vibrational eigenfrequencies $\omega_{0k}$ and vibrational eigenstates $\psi_k$ were determined numerically using the Fourier-grid Hamiltonian method[37]. This also yielded transition dipole moments by

numerically evaluating the integral $\langle \psi_0 | \mu | \psi_k \rangle$. The grids on which dipole moment and energies were evaluated were built from the single-point values by interpolation using cubic splines. The results of this procedure are shown in Supplementary Fig. 2. For the other normal modes, the values the harmonic vibrational frequencies and IR intensities (Supplementary Table 3) were scaled such that the mean frequencies/IR intensities for modes 14 and 15 match the mean frequencies/IR intensities calculated by the procedure described above and reported in Table 1 (gas phase, PCM).

The transition frequencies and transition dipole moments in solution were also obtained by studying static snapshots of $DMSO_2$ in water obtained via classical molecular dynamics simulations. The GROMACS package[38] was used with the OPLSAA forcefield[39] for $DMSO_2$ and TP3P[40] for water. The optimized geometry of $DMSO_2$ (see Supplementary Table 1) was placed at the center of a 30-Å box with 879 molecules of water and was equilibrated at a temperature of 298.15 K. During all MD runs, the coordinates of the central solute were kept frozen. From NVT trajectories with a total duration of 30 ns, snapshots of the solvent shell were extracted at a random time point every second 5 ps interval. All solvent molecules not within 7.5 Å of the solute molecule were removed. Into these solvent shells, $DMSO_2$ geometries were inserted in place of the central $DMSO_2$ molecule, where the atoms were displaced along the two relevant normal modes (see Supplementary Table 2). Vibrational frequencies and vibrational transition dipole moments were then evaluated as described in the previous paragraph and atoms in the solvent shell were explicitly included in the DFT calculation. The obtained frequency distributions for modes 14 and 15 are shown in Supplementary Fig. 3. The ensemble average over these distributions then yielded the central frequencies and transition dipole moments in Table 1 (explicit solvation).

To estimate the dephasing rate $\Gamma_k$ we conducted on-the-fly semi-classical dynamics simulations of $DMSO_2$ in water. A box was prepared containing one $DMSO_2$ molecule and 46 water molecules and was pre-equilibrated using classical molecular dynamics as described previously. The chosen force field was OPLSAA for $DMSO_2$ and TIP4P2005f[41] for water. The NPT pre-equilibration to 300 K, 1 bar yielded a box size of 11.7302 Å side length, corresponding to a density of 0.95 g/cm³. Then, on-the-fly semi-classical dynamics simulations using the BLYP-D3 functional and GTH pseudopotentials as implemented in the program package cp2k (version 6.1) were performed[42,43]. All calculations used four multigrids, a CUTOFF of 350 Ry and a REL_CUTOFF of 40 Ry. After NVT equilibration at 300 K for 11 ps, a 30 ps NVE trajectory was run with a time-step of 0.5 fs. The obtained geometries and electron densities were used to calculate the classical dipole autocorrelation function using the tessellation method by refs. 15, 16 as implemented in TRAVIS[44]. An estimate of $\Gamma_k$ was then obtained by fitting Lorentz functions to the Fourier transformed dipole autocorrelation function (Supplementary Fig. 4).

In summary, the ab initio model as presented in Figs. 2b, c and 3c uses the parameters displayed in Table 1 under explicit solvation for the transition frequencies and dipole moments, as well as the averaged estimate for the dephasing rate presented in Supplementary Fig. 4.

## Data availability

Source data for the figures in the main text are provided by the Source Data file. Additional information regarding the transient absorption setup, the quantum chemical calculations, and the theoretical model are provided by the Supplementary Information. Other data are available from the authors upon reasonable request.

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

## Acknowledgements

We thank Ferenc Krausz and Marinus Huber for useful discussions. This research was undertaken thanks, in part, to funding from the Technology Transfer Program of the Max Planck Society, and the Max Planck-UBC-UTokyo Center for Quantum Materials. R.d.V.-R. acknowledges funding by the German Research Foundation (DFG) under Germany's excellence strategy EXC 2089/1-39077620.

## Author contributions

M.T.P, M.H., T.B., R.d.V.-R., and I.P. designed the experiments and the theoretical framework. T.B. performed the experiments. T.B. and M.H. analyzed the experimental data. M.T.P. and D.K. performed the density functional theory simulations. M.H. and M.T.P. developed the model and analyzed the simulation data. M.T.P, M.H., and I.P. wrote the manuscript with input from all other authors. I.P. and R.d.V.-R. supervised the project.

## Funding

## Competing interests

The authors declare no competing interests.
