## [Peer Review File · Nature Communications]

Sub-optical-cycle light-matter energy transfer in molecular vibrational spectroscopyREVIEWER COMMENTS

Reviewer #1 (Remarks to the Author):

The paper describes interesting results of quantifying the emitted fields from a test sample, a compound featuring two coupled S=O modes (ss and as). The experiment is elegant and provides a textbook clarity of the interaction of radiation and a vibrational mode. The paper is not an easy read though, while enjoyable. I recommend it for publications given that the comments below are addressed.

Conclusion: the findings are not prioritized - it is not very clear what achievements of the study the authors are valued the most and how novel they are.

The authors claim that the molecular scale properties of the sample can be traced with precise measurements of emitted field. More specifics would be beneficial.

Line 99: It is unclear how the authors can attribute time-narrow oscillations around -100, 500 and 800 fs to different vibrational modes.

The authors use a whole paragraph (161-171) to discuss trivial effects of the solvent in computations with PCM solvents. This part should be shortened as it is not essential for the discussion.

The choice of CaF₂ (should be BaF₂ in the text) for producing a chirp in the pulse is strange as 5 mm CaF₂ absorbs most of radiation below 1150 cm⁻¹. (BaF₂ would be fine) It seems that in Fig. 3a some self modulation is apparent for the reference sample (black line). Could this modulation (if it is there) affect the results? If space allows, I would move the discussion of Fig. 4 to the text from the figure caption.

Minor suggestions

Line 60: The use of "Dirac-like excitation" forced me to look it up. I didn't find this terminology widely used. I suggest simple addition: "Dirac-like delta-function excitation" to help the readers to avoid the confusion. Impulsive regime is another accepted option.

Line 180: The work "complete" in the following "It is unclear insight into the complete coherent energy transfer dynamics" is misleading as it can be interpreted as 100% transfer. Rewarding is requested.

Line 109: "...energy is absorbed or emitted in a stepwise fashion with each half-cycle of the electric field." The periods of the switches observed do not match the half-cycle period (former is much smaller). The expression above suggests that there is a match of the two frequencies.

Experimental parameters, such as scanning range etc., are not given.

Reviewer #2 (Remarks to the Author):

This work provided a measurement of the absorption of the mid-infrared ultrafast broad laser pulse and the subsequent emission of IR light by the vibrating methylsulfonylmethane (DMSO₂) molecules in aqueous solution, with the electric-field-resolved spectroscopy with sub-optical-cycle temporal resolution. Furthermore, using chirped-pulse excitation (CPE), effects beyond the rotating wave approximation (RWA) were captured for the molecular coherence. The later discovery is significant and exciting, and this work represents a major advance and opens a new path for further investigation.

The data in this manuscript is reliable and support the conclusion and claims. The manuscript clearly presented the experimental and modelling details. The experimental methods are well established by the lead author's lab before.

It is therefore recommended towards acceptance for publication with some changes to address the questions below.

1. Only the data of 10mg/ml of DMSO₂ in water is presented. Is there a linear relationship regarding

to the DMSO₂ concentration in this range of concentration and significant absorbance? Figure 4e is calculation instead of results from experimental data, right? This may be important to exclude other possible nonlinear effects for the observed phenomenon.

2. What is the IR power dependence for the data in Figure 2? Is it also linear?
3. Line 94, CaF should be CaF₂, right?
4. Line 298, the FWHM cannot be 115cm⁻¹ in Figure 1a, right? Then, what is it?

Reviewer #1 (Remarks to the Author):

The paper describes interesting results of quantifying the emitted fields from a test sample, a compound featuring two coupled S=O modes (ss and as). The experiment is elegant and provides a textbook clarity of the interaction of radiation and a vibrational mode. The paper is not an easy read though, while enjoyable. I recommend it for publications given that the comments below are addressed.

R1.0: We thank the reviewer for the positive assessment of our work, recommendation to publish and constructive remarks, which we have addressed in the following.

Conclusion: the findings are not prioritized - it is not very clear what achievements of the study the authors are valued the most and how novel they are.

R1.1: We agree that the conclusions section was lacking emphasis in the aspects of measuring coherent transients and sub-cycle effects of the coherence (as also stressed by Reviewer 2), and an assessment of the novelty. We thank the reviewer for this comment and have adapted the conclusions section, as well as made a slight change to the "In this Letter..." section.

The authors claim that the molecular scale properties of the sample can be traced with precise measurements of emitted field. More specifics would be beneficial.

R1.2: The specific properties considered are the vibrational frequencies and transition dipole moments as well as their fluctuations due to the solvent environment, including screening. We have accordingly modified the relevant sentences in the abstract and the conclusions.

Line 99: It is unclear how the authors can attribute time-narrow oscillations around -100, 500 and 800 fs to different vibrational modes.

R1.3: Our model makes it very easy to assign features of CET(t) to specific vibrational modes. We can do that by only simulating CET(t) of one mode and observing which features remain. Since we are in the linear regime, the total CET(t) is the sum of the contributions of each mode. We have added an explanatory sentence to the text and marked the three time points -100 fs, 500 fs and 800 fs in Fig. 3c. We also marked 500 fs and 800 fs in Fig 4d, illustrating how these two maxima in CET(t) are caused by maximum coherence between the vibrational levels of the symmetric stretching vibration.

The authors use a whole paragraph (161-171) to discuss trivial effects of the solvent in computations with PCM solvents. This part should be shortened as it is not essential for the discussion.

R1.4: We thank the reviewer for this suggestion. To make the discussion more concise, we removed the paragraph by moving the presented data from the main text to a new Table 1 and only left a concluding sentence in the main text.

The choice of CaF₂ (should be CaF₂ in the text) for producing a chirp in the pulse is strange as 5 mm CaF₂ absorbs most of radiation below 1150 cm⁻¹. (BaF₂ would be fine)

R1.5: The reviewer is correct in that CaF₂ starts absorbing towards lower optical frequencies. For our spectrum, this results in a slight reduction of the full-width-half-maximum spectral bandwidth by 10% at the lower-frequency edge, consistent with transmission data from the supplier (Thorlabs). Because

we anyway measured two separate reference pulses, one for the FCE and one for the CPE case, and because there are no absorptive features in the truncated part of the spectrum (compare Fig. 3b), this is not relevant for the data presented in Fig. 3. We have replaced CaF by CaF₂ and added the incidence angle for the CaF₂ window in the supplementary information.

It seems that in Fig. 3a some self modulation is apparent for the reference sample (black line). Could this modulation (if it is there) affect the results?

R1.6: The modulation of the envelope of the black line in Fig. 3a does not stem from nonlinear effects in the CaF₂ window, because of the low intensity there ($I \sim 10^6$ W/cm², Kerr constant $n_2 \sim 10^{-16}$ cm²/W, see <https://doi.org/10.1364/OE.380702>). This modulation is due to the fact that the chirp maps out the shape of the (non-Gaussian) excitation spectrum to the time domain. Also, even if there were a nonlinear effect in the CaF₂ window, this would not affect the results: Because the window is placed before the liquid cuvette, this would just result in a moderately altered CPE, but would not compromise the consistency between the CPE reference and DMSO₂ sample measurements.

If space allows, I would move the discussion of Fig. 4 to the text from the figure caption.

R1.7: We have followed the suggestion of the reviewer.

Minor suggestions

Line 60: The use of “Dirac-like excitation” forced me to look it up. I didn’t find this terminology widely used. I suggest simple addition: “Dirac-like delta-function excitation” to help the readers to avoid the confusion. Impulsive regime is another accepted option.

R1.8: We thank the reviewer for the suggestion and have called it impulsive regime in all of the manuscript.

Line 180: The work “complete” in the following “It is unclear insight into the complete coherent energy transfer dynamics” is misleading as it can be interpreted as 100% transfer. Rewording is requested.

R1.9: We thank the reviewer for this comment and have changed the wording to remove the ambiguity.

Line 109: “...energy is absorbed or emitted in a stepwise fashion with each half-cycle of the electric field.” The periods of the switches observed do not match the half-cycle period (former is much smaller). The expression above suggests that there is a match of the two frequencies.

R1.10: In the quoted paragraph we discuss two effects, the coherent transients on a larger timescale and the sub-cycle oscillations. This specific sentence refers to the latter. The frequency of the sub-cycle oscillations does indeed match the half-cycle period of the electric field (ca. 15 fs). For clarification, we have changed the zoom factor of Fig. 3d and added vertical lines that illustrate how the sub-cycle dynamics line up with the zeros of the electric field. We also changed the wording to more clearly distinguish between the coherent transients and the sub-cycle oscillations.

Experimental parameters, such as scanning range etc., are not given.

R1.11: We have added the scanning range to the supplementary information, and also mentioned the total measurement time, focus size in the sample cell, EOS crystal orientation, chopper frequency, and short-pass filter wavelength.

Reviewer #2 (Remarks to the Author):

This work provided a measurement of the absorption of the mid-infrared ultrafast broad laser pulse and the subsequent emission of IR light by the vibrating methylsulfonylmethane (DMSO₂) molecules in aqueous solution, with the electric-field-resolved spectroscopy with sub-optical-cycle temporal resolution. Furthermore, using chirped-pulse excitation (CPE), effects beyond the rotating wave approximation (RWA) were captured for the molecular coherence. The later discovery is significant and exciting, and this work represents a major advance and opens a new path for further investigation.

The data in this manuscript is reliable and support the conclusion and claims. The manuscript clearly presented the experimental and modelling details. The experimental methods are well established by the lead author's lab before. It is therefore recommended towards acceptance for publication with some changes to address the questions below.

R2.0: We thank the reviewer for the positive overall assessment of our work and its significance. In the following we address all of the reviewer's concerns in detail.

1. Only the data of 10mg/ml of DMSO₂ in water is presented. Is there a linear relationship regarding to the DMSO₂ concentration in this range of concentration and significant absorbance? Figure 4e is calculation instead of results from experimental data, right? This may be important to exclude other possible nonlinear effects for the observed phenomenon.

R2.1: We thank the reviewer for this observation. Indeed, at very high concentration the change of the field as a consequence of the interaction with the molecules is not linear with the concentration anymore. However, our numerical model used for Figs. 1-4 includes this effect by using the Lambert-Beer formula for H₂O instead of its Taylor approximation (see "Role of the reference measurement" section in the supplementary information). The description of Fig. 4e (moved from the caption to the main text in response to R1.7) refers to a linear and quadratic scaling behaviour at small concentrations. We previously had not quantified up to which concentrations this scaling remains valid, and have consequently added Figure S5 to the supplementary information, showing that the deviation at 10 mg/ml is still smaller than 14%.

Concerning the other question: The reviewer correctly notes that Figure 4e shows results from ab-initio simulations and not from experimental data. As a result, our observations (quadratic scaling of the reemission, increase of the reemission due to the environment) are not to be attributed to unexpected nonlinear behaviours in the measurement data (e.g. with respect to the concentration, or with respect to the IR power, see R2.2).

2. What is the IR power dependence for the data in Figure 2? Is it also linear?

R2.2: The linearity of the EOS traces with respect to the impinging power was confirmed with a very similar setup in <https://doi.org/10.1021/acs.analchem.9b05744> (Figure 4) by comparison to FTIR data acquired at much lower intensity (with a thermal IR source). We have added this reference to the manuscript.

3. Line 94, CaF should be CaF₂, right?

R2.3: We thank the reviewer for noting this and have fixed this.

4. Line 298, the FWHM cannot be 115cm^{-1} in Figure 1a, right? Then, what is it?

R2.4: We thank the reviewer for noting this mistake. The given FWHM was wrong by a factor of 2π and should be 18.3 cm^{-1} . We have changed both the figure and the description accordingly.

Further small changes:

- Caption of Fig 4: we fixed a typo (asymmetric \rightarrow symmetric)
- We have improved the explanation of the data without environment, PCM, and explicit solvation.
- Supplementary information: We have renamed the section “Scaling behaviour of CET(t) with concentration” to “Scaling behaviour of the absorbed and reemitted energy with concentration”

REVIEWER COMMENTS

Reviewer #1 (Remarks to the Author):

The authors addressed all my concerns. I recommend the paper for publication.

Reviewer #2 (Remarks to the Author):

The revised manuscript addressd the comments and questions. It is recommended to accept for publication in current form.

Reviewer #1 (Remarks to the Author):

The authors addressed all my concerns. I recommend the paper for publication.

Response: We thank the reviewer for the recommendation and the helpful comments during the review process.

Reviewer #2 (Remarks to the Author):

The revised manuscript addressd the comments and questions. It is recommended to accept for publication in current form.

Response: We thank the reviewer for the recommendation and the helpful comments during the review process.